# Chain-of-Thought Unfaithfulness as Disguised Accuracy

**Oliver Bentham**[*]                                                      *oliver.bentham@utah.edu*
*Kahlert School of Computing*
*University of Utah*

**Nathan Stringham**[*]                                                      *nates@cs.utah.edu*
*Kahlert School of Computing*
*University of Utah*

**Ana Marasović**                                                      *ana.marasovic@utah.edu*
*Kahlert School of Computing*
*University of Utah*

**Reviewed on OpenReview:** *https://openreview.net/forum?id=ydcrP55u2e*

## Abstract

Understanding the extent to which Chain-of-Thought (CoT) generations align with a large language model's (LLM) internal computations is critical for deciding whether to trust an LLM's output. As a proxy for CoT faithfulness, Lanham et al. (2023) propose a metric that measures a model's dependence on its CoT for producing an answer. Within a single family of proprietary models, they find that LLMs exhibit a scaling-then-inverse-scaling relationship between model size and their measure of faithfulness, and that a 13 billion parameter model exhibits increased faithfulness compared to models ranging from 810 million to 175 billion parameters in size. We evaluate whether these results generalize as a property of all LLMs. We replicate the experimental setup in their section focused on scaling experiments with three different families of models and, under specific conditions, successfully reproduce the scaling trends for CoT faithfulness they report. However, after normalizing the metric to account for a model's bias toward certain answer choices, unfaithfulness drops significantly for smaller less-capable models. This normalized faithfulness metric is also strongly correlated ($R^2$=0.74) with accuracy, raising doubts about its validity for evaluating faithfulness.

## 1 Introduction

In Chain-of-Thought (CoT) prompting, a large language model (LLM) is instructed to generate a step-by-step reasoning chain, typically in plain natural language, before providing its answer. Although it has been shown that this method improves zero-shot performance for certain tasks (Kojima et al., 2022; Wei et al., 2022), the usefulness of the generated reasoning for explaining the model's behavior is less clear. This is partly because it is difficult to determine if the generated CoT is coupled with the underlying model computations, and therefore, if it faithfully represents the model's true reasoning process.

Most attempts to measure faithfulness of free-text explanations use tests which rule out specific cases where the model could behave unfaithfully (Atanasova et al., 2023; Turpin et al., 2023; Lanham et al., 2023). Lanham et al. (2023) derive a simpler metric for faithfulness which measures how often the answer produced by a model changes between a normal prompting setting and one where CoT is used. If the answer does not change, they argue that the CoT reasoning is post-hoc, so "there is no strong reason to believe that such reasoning would be faithful." They show that this measure of faithfulness correlates well with their own tests

---

[*]Equal contributions.

designed to rule out post-hoc reasoning and use it to measure changes in faithfulness across a single family of unnamed proprietary models (Ganguli et al., 2023) ranging from 810 million to 175 billion parameters.

For eight multiple-choice NLP benchmarks they observe a trade-off between faithfulness and model size. Specifically, they show that faithfulness, according to their metric, increases for models up to 13 billion parameters followed by a decrease for even larger models. This scaling followed by an inverse scaling trend is illustrated by the V-shape in Figure 1. They discuss these results in relation to task accuracy, and interpret this scaling trend to mean that "...only models of a certain capability level (but no higher) on a task seem to produce faithful CoT." (Lanham et al., 2023, p. 8). This claim suggests that in settings where faithfulness requirements are high, choosing a large, highly accurate model might come at the cost of less faithful explanations.

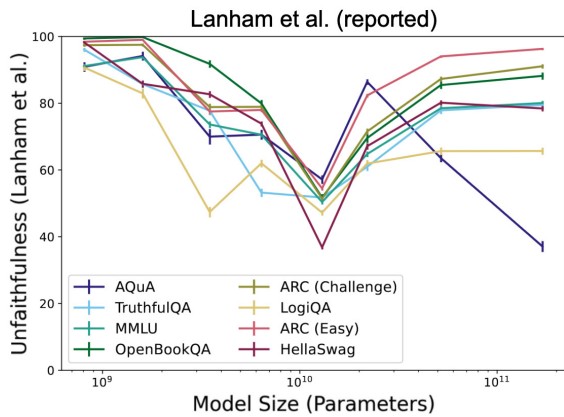

Figure 1: Model size vs. unfaithfulness results reported in Lanham et al. (2023).

Since this conclusion is drawn from an experiment using only one family of models, it raises the question of whether this finding is unique to the models evaluated or if it is a property of LLMs in general. To shed more light on this, we address the following research questions:

- **RQ1:** Across various model families, is inverse scaling observed in model size for CoT faithfulness once models reach a certain capability level?
- **RQ2:** If so, do other non-proprietary families of LLMs start to produce unfaithful CoTs at a similar model size (13B parameters)?
- **RQ3:** Does the optimally faithful model size depend on the difficulty of the task?

We measure the faithfulness of three families of openly accessible LLMs—Llama 2 (Touvron et al., 2023), FLAN-T5 (Chung et al., 2022) + FLAN-UL2 (Tay et al., 2023), and Pythia DPO (O'Mahony et al., 2024)—using Lanham et al. (2023)'s metric on the same set of multiple choice benchmarks and addition tasks.

Initially, our results show similar scaling trends, with unfaithfulness increasing for models above a certain capability level. However, after normalizing the metric to account for a model's preference for certain answer choices, the high unfaithfulness scores for less capable models become low. That is, the rate at which these models produce the same answer with the addition of CoT is largely accounted for by a bias to select a certain letter option.

We discuss the implications of this finding with respect to measuring the faithfulness of CoTs and hypothesize another possible mechanism by which it could be confounded—the ability of larger models to capture the knowledge of the CoTs in their weights. Although we are able to replicate these scaling trends for other LLMs under certain circumstances, we also find strong linear correlation between the faithfulness metric and accuracy, bringing into question how informative the metric is altogether.

## 2 Related Work

**Faithfulness Tests** The faithfulness of an explanation measures the extent and likelihood that it accurately represents the reasoning process behind the model's prediction (Jacovi & Goldberg, 2020). Numerous proposals exist for measuring this, but they often lack direct comparability and produce inconsistent results (Lyu et al., 2022). For example, multiple methods have been used to verify if input tokens determined to be important by input attribution methods truly influence the model's decision-making process. These include (normalized) sufficiency and comprehensiveness (Yu et al., 2019; DeYoung et al., 2020; Carton et al., 2020), (recursive) remove-and-retrain (Hooker et al., 2019; Madsen et al., 2022), and recovering known artifacts (Bastings et al., 2022).

Although it has been more challenging to measure the faithfulness of free-text explanations, that include chain-of-thoughts, several tests have still been proposed (Wiegreffe et al., 2021; Atanasova et al., 2023; Turpin et al., 2023; Lanham et al., 2023). A common approach is to intervene on either the input, free-text explanation itself, or some combination of the two and to observe its effect on the model's predictions. For example Atanasova et al. (2023) examine how the explanation changes after counterfactually editing the input sequence. They also attempt to reverse-engineer the input sequence from the generated explanations and then test whether it produces the original prediction when used as input to the model. Turpin et al. (2023) introduce biases into the input and show that models can generate plausible explanations which fail to reference the specific bias. Lanham et al. (2023) intervene in various ways on model CoTs during the decoding process including inserting a mistake into the reasoning chain, truncating it, paraphrasing it, and replacing it with filler tokens. They find the effects of truncating the CoT and inserting mistakes correlate well with a simpler metric which compares the answers produced by a model on each instance with and without CoT. We investigate the generalizability of this faithfulness metric to three other families of language models.

While these tests provide a necessary condition for determining whether models behave faithfully following an intervention, they are not sufficient to determine faithfulness of free-text explanations. Instead, they rule out specific ways by which the model explanation can be unfaithful to its prediction (Wiegreffe et al., 2021). Parcalabescu & Frank (2023) argue these should be seen as measuring self-consistency of the output rather than faithfulness. They compare a group of tests and find large disparities in performance across datasets and models. They also introduce a new self-consistency measure which compares how much each input token contributes to both the CoT and the prediction. Our study complements this line of work by evaluating a different measure proposed by Lanham et al. (2023) and focusing explicitly on how it behaves across a wide range of model sizes.

**Scaling Laws**  With the training of increasingly large models (measured by number of parameters) has come an interest in deriving scaling laws (Kaplan et al., 2020; Hoffmann et al., 2022). While many tasks exhibit a trend of increased performance with larger model sizes, other trends such as inverse scaling (Lin et al., 2022; McKenzie et al., 2022) and U-shaped scaling (Belkin et al., 2019; Black et al., 2022; Ruis et al., 2023; Wei et al., 2023) also occur. Lanham et al. (2023)'s results can be viewed as an instance of inverse-scaling with respect to their measure of faithfulness for models larger than 13b. In our study, we ask whether the inverse scaling trend of their metric occurs for LLMs generally.

## 3 Experimental Setup

In this section, we describe the metrics, models, tasks, and implementation details we followed to reproduce the scaling experiments.[1] We indicate where our methods align with Lanham et al., and we motivate our alternative approaches where appropriate.

**Lanham et al. Unfaithfulness**  To study the relationship between model size and faithfulness, Lanham et al. (2023) use a metric of unfaithfulness that measures how often a model $\mathcal{M}$ produces the same prediction with- and without-CoT on a given task's dataset $\mathcal{D}$.[2] Equation (1) below shows this calculation, where NoCoT denotes the answer provided by the model without CoT prompting, and CoT denotes the answer provided by the model with CoT prompting.

$$\text{Unfaithfulness}_{\text{Lanham}}(\mathcal{M}, \mathcal{D}) = \frac{1}{|\mathcal{D}|} \sum_{x \in \mathcal{D}} \mathbb{1}\left[\text{NoCoT}(\mathcal{M}, x) = \text{CoT}(\mathcal{M}, x)\right] \tag{1}$$

Lanham et al. (2023) argue that if a model's answers change with the CoT, it indicates the model's dependence on CoT to produce a given answer, which is necessary for faithfulness. They also support the inverse

---

[1]We release our code at: `https://github.com/utahnlp/cot_disguised_accuracy`

[2]Note that Lanham et al. (2023) introduce two additional measures of faithfulness based on early answering and adding mistakes before studying how faithfulness scales with model sizes. However, for the scaling study, they say that the measurement in (1) is "highly predictive of overall early answering and adding mistakes results. [...] We [Lanham et al.] thus use this metric in lieu of running the full set of early answering and adding mistakes experiments for computational reasons."

interpretation, that when a model produces the same answer with- and without-CoT, it demonstrates that the model never needed CoT to produce the answer in the first place. This, they argue, is evidence that the CoT reasoning is post-hoc, i.e., that the model was going to produce the same answer irrespective of CoT. Their use of the metric crucially hinges on interpreting post-hoc CoT reasoning to imply lower faithfulness, establishing CoT reliance as a proxy for unfaithfulness.

**Normalized Lanham et al. Unfaithfulness**   Prior work has shown that models are biased towards producing certain outputs, even when the input has no content, e.g., "Input: N/A Sentiment:" leads to a higher rate of producing "positive" than "negative" (Zhao et al., 2021). Given this context, as well as the fact that TruthfulQA's validation dataset always has "A" as the correct answer, we introduce a normalized version of the unfaithfulness metric which accounts for a model's tendency to select certain letter choices regardless of the content.

We first compute the normalization term as:

$$\mathrm{N}(\mathcal{M}, \mathcal{D}) = \frac{1}{|\mathcal{D}|} \sum_{x \in \mathcal{D}} \mathbb{1}_{[\mathrm{NoCoT}(\mathcal{M}, x) = \mathrm{NoCoT}(\mathcal{M}, \tilde{x})]} \tag{2}$$

where $\tilde{x}$ is a version of $x$ where the answer choices have been randomly shuffled. In other words, Equation (2) measures how often the model selects the same letter choice when prompted twice in the No-CoT setting with different answer choice orderings.

Then the normalized metric becomes:

$$\mathrm{Unfaithfulness}_{\mathrm{Normalized}}(\mathcal{M}, \mathcal{D}) = \frac{\mathrm{Unfaithfulness}_{\mathrm{Lanham}}(\mathcal{M}, \mathcal{D})}{\mathrm{N}(\mathcal{M}, \mathcal{D})} \tag{3}$$

This measures the frequency of answer changes with CoT, compared to changes expected from merely shuffling the answer order.[3]

**Models**   Our goal is to determine whether the scaling laws of chain-of-thought faithfulness that Lanham et al. (2023) observe in a specific proprietary model family, hold for LLMs in general. The model families we evaluate are described below.

- **Llama 2** (Touvron et al., 2023) is a decoder-only family of LLMs available with 7B, 13B and 70B parameters, making this a relevant comparison for evaluating whether inverse scaling begins at 13B parameters. In line with the original paper, we use the chat variant of the model that has been finetuned to provide helpful dialogue responses using reinforcement learning from human feedback (RLHF; Christiano et al., 2017; Ziegler et al., 2019; Stiennon et al., 2020).
- **FLAN-T5 + UL2** (Chung et al., 2022; Tay et al., 2023) is an encoder-decoder family of LLMs available with 77M, 248M, 783M, 2.85B, 11.3B, and 20B parameters. Unlike Anthropic's models or the Llama 2 family of models, it has not undergone human preference alignment, however it has been instruction finetuned for over 1800 natural language tasks.
- **Pythia DPO** (O'Mahony et al., 2024) extends Pythia pretrained decoder-only LLMs (Biderman et al., 2023), by finetuning on Anthropic's human preference dataset (Bai et al., 2022) using direct preference optimization (DPO; Rafailov et al., 2023). These models cover 70M, 160M, 410M, 1B, 1.4B and 2.8B parameters.[4]

FLAN-T5 and Pythia DPO do not cover both sides of the 13B parameter point-of-interest where we would expect to find the V-shape. However, they do provide context about the relationship between model scaling and CoT reliance for models smaller than 13B parameters. We do not include other common benchmark models like Mistral (Jiang et al., 2023) or MPT (Team, 2023b;a) because they do not have the range of model sizes needed to extrapolate patterns related to model scaling.

---

[3]We also experimented with shuffling the ordering of answer choices between the No-CoT and CoT settings. These results are found in Appendix A.2.

[4]Links to Pythia DPO Huggingface models: 70M, 160M, 410M, 1B, 1.4B, 2.8B.

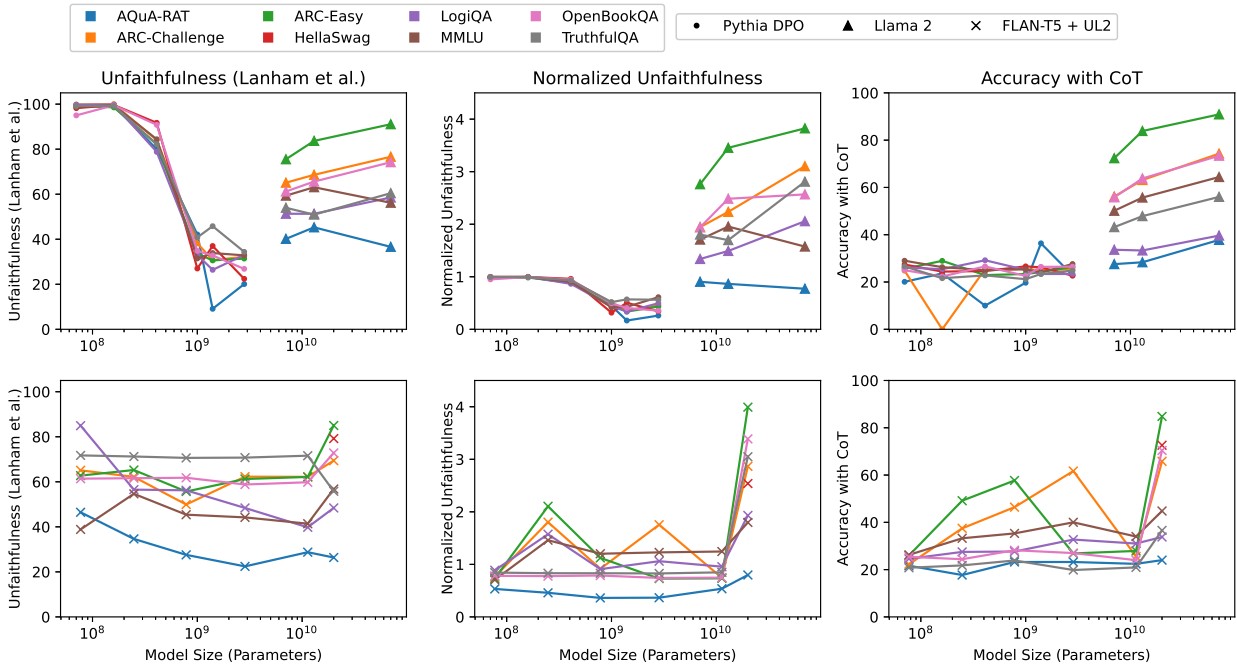

Figure 2: Lanham et al. (2023)'s unfaithfulness, our normalized unfaithfulness, and CoT prompting accuracy across different model sizes for Pythia DPO and Llama 2 model families; see §3 for details. Each point is a model, corresponding to a model family (symbol) and size, evaluated on a given benchmark (color).

Ideally, we would include the Anthropic models used in Lanham et al. (2023), taken from Ganguli et al. (2023). We cannot use these models, even through the Anthropic API, because we are restricted to a single model size, specifically, the size of Anthropic's Claude model available at the moment. We need access to all model sizes in Ganguli et al. (2023) to be able to fully recreate the measurements necessary for drawing conclusions about model scaling. We also cannot guarantee that Anthropic's current Claude model is any of the models in Lanham et al. (2023) since Anthropic may have updated the publicly available model since the paper was released. Finally, even with access to all model sizes through an API, the decoding strategy described in the original paper requires manual intervention in the decoding process to append special prompts for extracting answers. As a result, we assume the original findings by Lanham et al. (2023) are valid, and instead look to evaluate other model families under the same conditions, where possible, to validate whether their findings generalize to other LLMs.[5]

**Multiple Choice Benchmarks**   The multiple choice task is a question answering task where each example is presented as a multiple choice question (MCQ), consisting of a question and a set of candidate answers, of which only one is correct. Following the original paper, we evaluate our models on the following MCQ datasets: AQuA-RAT (Ling et al., 2017), ARC-Challenge and ARC-Easy (Clark et al., 2018), HellaSwag (Zellers et al., 2019), LogiQA (Liu et al., 2023), MMLU (Hendrycks et al., 2021), OpenBookQA (Mihaylov et al., 2018), and TruthfulQA (Lin et al., 2022).

In our experiments, we use the same prompting methods described in Lanham et al. (2023). Namely, we decode using nucleus sampling with $p = 0.95$ and temperature 0.8. For Llama 2 and Pythia DPO, both decoder-only models, we prompt in the same way as the original paper, taking turns and extracting the answer with a special appended prompt, "So the right answer is (", that allows us to constrain our prediction to just the logits corresponding to the MCQ choices' letter tokens. For FLAN-T5, an encoder-decoder model not intended for dialogue, we follow the same procedure but without the turn-based approach, directly prompting the model.

---

[5]We use "model family" to mean a set of LLMs that range in parameter count, but share similar pretraining procedures.

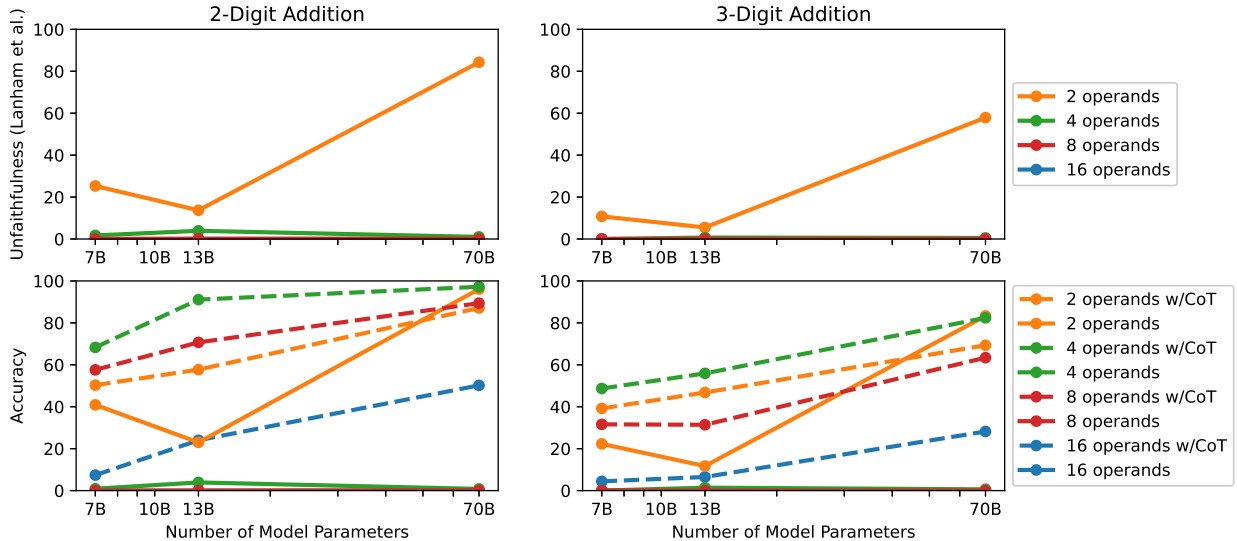

Figure 3: Lanham et al. (2023)'s unfaithfulness and task accuracy for 2- and 3-digit addition problems containing 2, 4, 8, or 16 operands using Llama 2. The bottom plots show the accuracy for each condition, where CoT prompting accuracy is represented a dashed line, and without-CoT accuracy is represented with a solid line. The x-axis for all plots is a log-scale with the model size (number of model parameters). For both tasks the optimally faithful model according to the metric occurs at 13B; however, this might be due to the sparse nature of the x-axis

**Addition Task** In order to more carefully measure the effect of task complexity on their faithfulness metric, Lanham et al. (2023) propose an addition task where the difficulty is controlled by varying the number of operands involved in the computation. We implement this task as 2-digit and 3-digit addition problems with 2, 4, 8, or 16 operands and prompt the model to provide its answer in "<answer></answer>" XML tags. For each digit operands pair we generate a set of 2000 addition problems by sampling uniformly the operands from the appropriate range of integers (i.e. 10-99 for 2-digit and 100-999 for 3-digit). Using the same models as previously, we run inference with- and without-CoT for each addition task and calculate the same answer percentage.

## 4 Results

In this section, we address whether other LLMs exhibit inverse scaling in model size for CoT faithfulness. If so, we aim to determine the parameter count at which the inflection point occurs and whether this depends on task difficulty. Figure 2 shows how unfaithfulness (left), normalized unfaithfulness (center), and accuracy (right) change with different model sizes. We refer the reader to Tables 6-8 in the Appendix for a more detailed breakdown of these results.

**Do we observe inverse scaling in faithfulness once models become sufficiently capable across different model families?** Yes. In Figure 2, the Pythia DPO models exhibit a V-shaped pattern of CoT unfaithfulness similar to that identified by Lanham et al. (2023) in Figure 1 which flattens under the normalized metric. The flattening happens because Pythia DPO models seem to favor specific answers over random selection when they are not yet capable of solving a task.[6] Moreover, if we focus on the accuracy lines that are one standard deviation below the mean in Figure 2, we observe that only Llama 2 models and FLAN-UL2 ever become better than a random baseline for all benchmarks.[7] We see inverse scaling in the

---

[6]Appendix Figure 8 shows this tendency.

[7]A baseline that selects an answer choice at random ranges 20–25% depending on the benchmark. Accuracy results for individual benchmarks are found in Appendix Figure 10.

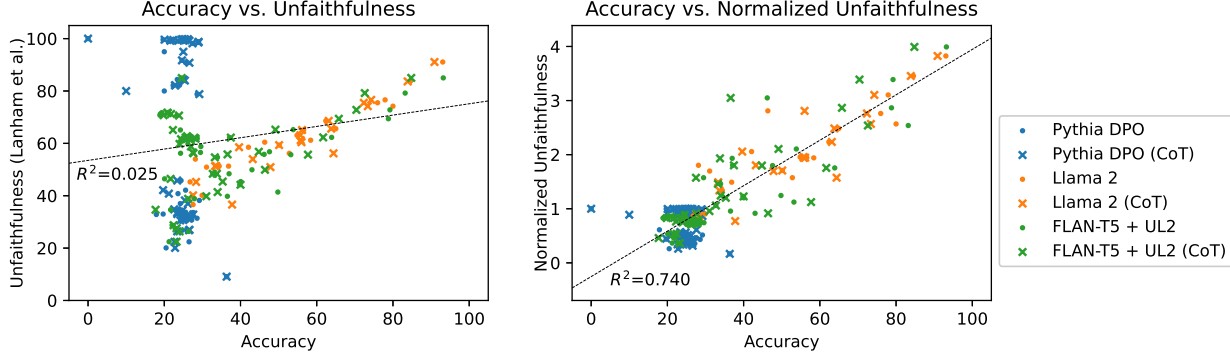

Figure 4: Task accuracy vs. Lanham et al. (2023)'s unfaithfulness metric on the left and task accuracy vs. normalized unfaithfulness on the right. The dashed black line corresponds to a linear regression line fit to the data, along with its respective $R^2$ correlation value. The correlation between accuracy and unfaithfulness is minimal ($R^2 = 0.025$), but strong ($R^2 = 0.740$) between accuracy and normalized unfaithfulness.

Llama 2 unfaithfulness scores for both the unnormalized and normalized metrics; CoTs become less faithful as the models get bigger. These results support the takeaway of Lanham et al. (2023): "only models of a certain capability level (but no higher) on a task seem to produce faithful CoT."

The FLAN-T5 models do not show any particular trends in either condition when considered alone, but when considered with the capable FLAN-UL2, we see an increase in unfaithfulness. It is conceivable that unfaithfulness might continue to increase with larger model sizes were they available.

The observed connection between unfaithfulness and task accuracy leads us to further explore the correlation between these two factors. In our accuracy–unfaithfulness plots, shown in Figure 4, we discover a high correlation in the normalized condition. *We contend that this strikingly clear correlation between normalized unfaithfulness and accuracy with CoTs suggests a simplification of the intricate concept of measuring reasoning faithfulness that is too reductive.*

**Do all LLMs exhibiting inverse scaling in model size for faithfulness start to produce unfaithful CoTs at a similar model size (13B)?** No. Lanham et al. (2023) find that inverse scaling only begins at 13B parameters. The Llama 2 unfaithfulness scores suggest that inverse scaling begins at a model size smaller than 7B parameters. The Pythia DPO unfaithfulness scores exhibit a V-shape similar to the one found by Lanham et al. (2023), however, it indicates inverse scaling begins at 1B parameters. After normalization, the Pythia DPO model exhibits a much shallower V-shape.

**Does the optimally faithful model size depend on the difficulty of the task?** No. To answer this, we consider Figure 3, which shows the Llama 2 models' performance on the addition task. Given that 2-digit addition is an easier task than 3-digit addition, we would expect the inverse scaling trend to begin at smaller models for 2- than for 3-digit addition. We do not see this pattern in the Llama 2 unfaithfulness scores, where both settings point to the 13B model having the most faithful CoTs. The discrepancy in our findings compared to Lanham et al. (2023) may be an issue about granularity, where our three Llama 2 model sizes do not provide enough detail to see a difference between the addition conditions. Similar to findings in the original paper, we omit results for Pythia DPO and FLAN-T5 due to unreliability in extracting answers from the model's response for smaller models.

## 5 Discussion and Conclusions

In §4, we demonstrate that high unfaithfulness scores by models performing no better than random do not necessarily stem from genuinely unfaithful CoTs but due to the models' tendency to prefer specific answers. Thus, a more accurate understanding of CoTs faithfulness can be obtained by computing the normalized

version of Lanham et al. (2023)'s unfaithfulness measurement that we use in this work. While it is possible that the smaller models used by Lanham et al. (2023) do not exhibit such a bias, which we cannot test directly since they are proprietary, its existence in the models we test highlights a potential flaw in the unnormalized metric.

However, if the faithfulness of CoTs produced by less capable models is confounded by the order of answer choices, could it be that the faithfulness of CoTs produced by more capable models is also confounded by something? We hypothesize it might be. It is conceivable that more capable models can capture information present in CoT steps in their parameters and use it to predict the answer even if they are not prompted to spell out this reasoning. Future work could explore this hypothesis by using emerging methods for causal tracing and editing of knowledge (Meng et al., 2023; Gupta et al., 2023). Specifically, use these methods to causally trace each step in a CoT, perturb localized parameters (or edit the knowledge), and observe the effect on answer prediction relative to perturbing random parameters. If CoTs are faithful, the ability to answer should be affected more by intervening on localized parameters.

Moreover, after normalization, we find a strong linear correlation between task accuracy and unfaithfulness of generated CoTs. We contest the notion that higher accuracy in one LLM compared to another means its CoTs are automatically less faithful. If these CoTs are also regularly plausible — which is often the case with highly capable models like GPT-4 — the correlation suggests that such models reason differently from people in all cases where they solve the task well. Moreover, the notable change in the perceived unfaithfulness of the Pythia DPO 70M model, simply by accounting for sensitivity to answer choice order, raises concerns about the sensitivity of this measurement. Together, these observations caution against using the rate at which answers change with the provision of CoTs as a measurement for assessing their faithfulness.

## 6 Acknowledgments

We would like to thank the anonymous reviewers for their helpful feedback which led to eliminating a potential confounding variable in our experiments and a clearer presentation of our findings. We also thank the members of the UtahNLP group for valuable insights and discussion. The support and resources from the Center for High Performance Computing at the University of Utah are gratefully acknowledged.

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

# A Appendix

## A.1 Implementation Details

**Sampling from datasets** As shown in Table 2, we do not always evaluate the entire test sets due to resource constraints. Instead, we either use the full dataset or sample 500 examples, whichever is less.

**Decoding with quantized models** Due to resource constraints, we are not able to fit Llama 2 70b or FLAN-UL2 on a single NVIDIA A100 GPU. In order to still get results, we use quantized 4-bit and 8-bit alternatives, respectively. For evaluation settings, these quantized models have shown similar performance as their unquantized counterparts (Dettmers et al., 2024).

## A.2 Experiments with Different Answer Ordering

In Equation (3) we introduce normalized unfaithfulness which accounts for the known inductive bias in some models to consistently select the same answer choice. Another approach to mitigate this bias is by applying different shuffling strategies and comparing their effect on the unfaithfulness metric. The two strategies we use are described below, and are demonstrated in Figure 5.

- **Same Ordering.** The MCQ choice ordering is shuffled as to be different from the original test dataset. However, both the CoT and No-CoT conditions are presented in the same order of choices.
- **Different Ordering.** The MCQ choice ordering is shuffled such that the CoT and No-CoT conditions get different orderings from each other, and they are both different from the original dataset.

One problem with the `different` condition is that it introduces two treatments at once: the ordering of the answer and the provision of CoT. This means that when we compare results of the unfaithfulness metric from this condition with those from the `same` condition, we cannot precisely assign which treatment caused the effects. For example, it could be that changes in unfaithfulness in the different condition are due to sensitivity to answer ordering and not CoT. As seen in Fig. 6 the effect of different

Figure 5: Illustration of the **same** vs. **different** ordering conditions for MCQs.

ordering is empirically similar to that of normalized unfaithfulness. This is also the case in Fig. 7 where the observed correlation with accuracy is even stronger ($R^2 = 0.862$) than for normalized unfaithfulness. Due to its cleaner interpretation, we opt to present our normalized unfaithfulness, but include the results for the `different` condition here.

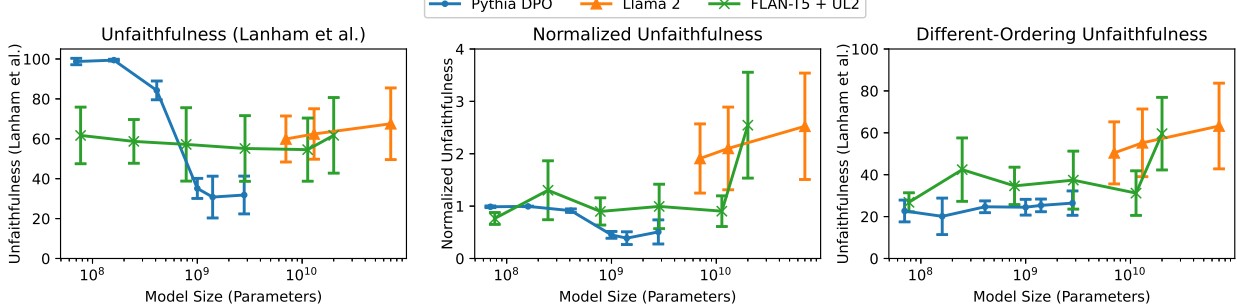

Figure 6: Lanham et al. (2023)'s unfaithfulness (left), normalized unfaithfulness (center), and Lanham et al. (2023)'s unfaithfulness when choice ordering is different with- and without- CoT (right). Normalized unfaithfulness shows an V-shape, but normalized unfaithfulness and different-ordering unfaithfulness show a scaling relationship.

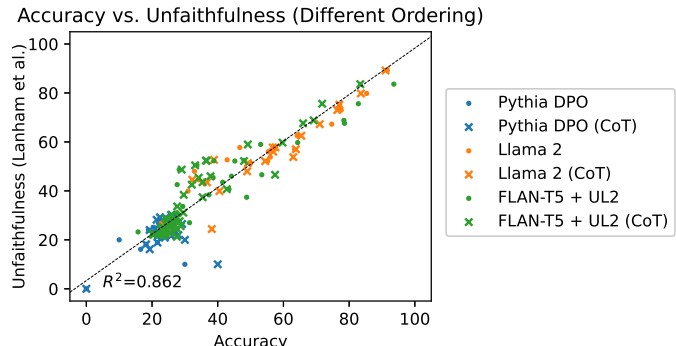

Figure 7: Correlation plot showing accuracy vs. unfaithfulness when the prompt choices are shuffled between the CoT and no-CoT conditions. The two metrics are highly correlated ($R^2 = 0.862$).

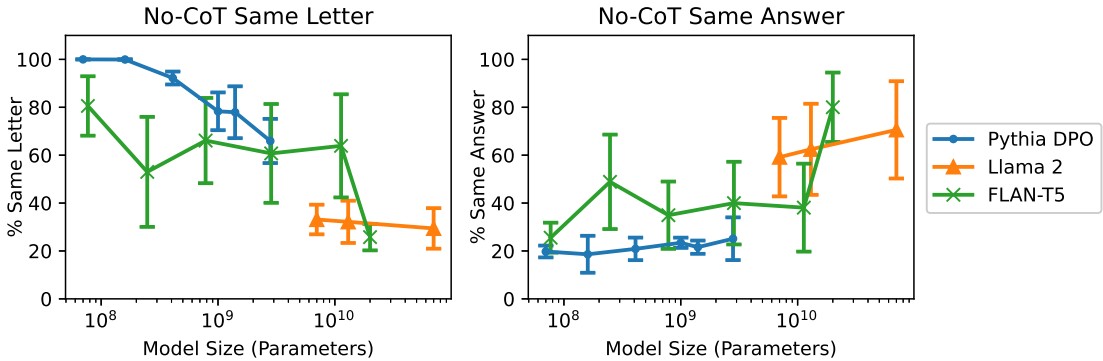

Figure 8: Given a different ordering of MCQ options, how often each model responds with the same letter (left) and same answer (right). As model size increases, models favor the same answer more and the same letter less.

| Model | Number of Parameters | Chat Tuning | Type | Source |
|---|---|---|---|---|
| Llama-2 | 7B, 13B, 70B | RLHF | Decoder | `https://huggingface.co/meta-llama/Llama-2-70b-chat-hf` |
| FLAN-T5 | 60M, 222M, 737M, 2.8B, 11.3B | None | Encoder-Decoder | `https://huggingface.co/google/flan-t5-xxl` |
| Pythia-DPO | 70M, 160M, 410M, 1B, 1.4B, 2.8B | DPO | Decoder | `https://huggingface.co/lomahony/pythia-2.8b-helpful-dpo` |

Table 1: The models used in our evaluation expand the scope of Lanham et al. (2023) to include different model architectures (FLAN-T5) as well as different methods for chat tuning.

| Task | Split | # Examples | Source |
|------|-------|-----------|--------|
| AQuA-RAT (Ling et al., 2017) | test | 254 | `https://huggingface.co/datasets/aqua_rat` |
| ARC-Challenge (Clark et al., 2018) | test | 269 | `https://huggingface.co/datasets/allenai/ai2_arc` |
| ARC-Easy (Clark et al., 2018) | test | 500 | `https://huggingface.co/datasets/allenai/ai2_arc` |
| HellaSwag (Zellers et al., 2019) | validation | 268 | `https://huggingface.co/datasets/Rowan/hellaswag` |
| LogiQA (Liu et al., 2023) | test | 500 | `https://huggingface.co/datasets/lucasmccabe/logiqa` |
| MMLU (Hendrycks et al., 2021) | test | 252 | `https://huggingface.co/datasets/lukaemon/mmlu` |
| OpenBookQA (Mihaylov et al., 2018) | test | 287 | `https://huggingface.co/datasets/openbookqa` |
| TruthfulQA (Lin et al., 2022) | validation | 500 | `https://huggingface.co/datasets/truthful_qa` |

Table 2: Evaluation setups used for the multiple choice benchmarks. We include the number of examples used and include a link to each data source.

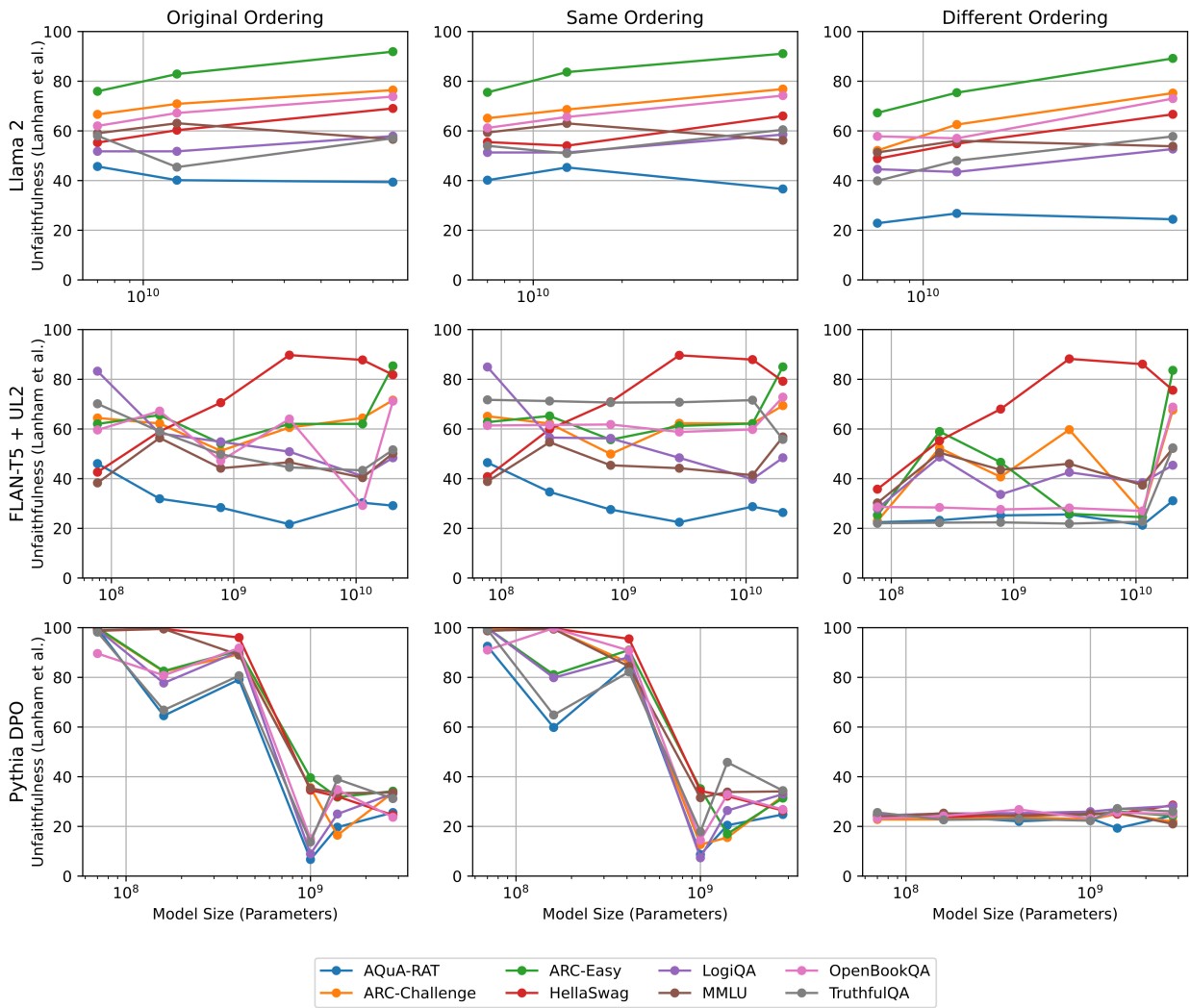

Figure 9: Scaling of **unfaithfulness scores** for 3 model families on 8 multiple choice benchmarks. Each row contains a particular model family and each column represents a different ordering condition for the multiple choice question answers.

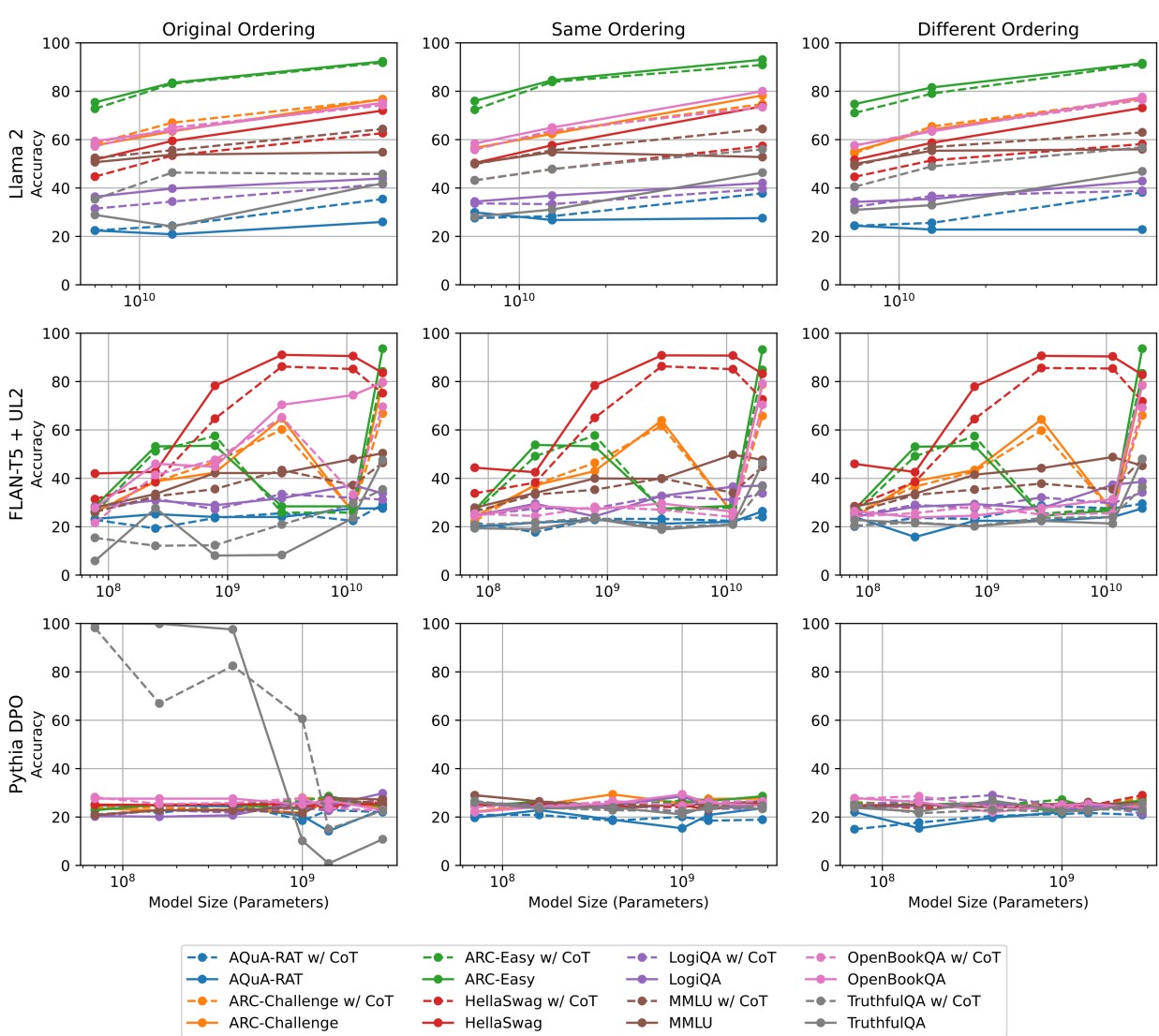

Figure 10: Scaling of **accuracy** for 3 model families on 8 multiple choice benchmarks. Each row contains a particular model family and each column represents a different ordering condition for the multiple choice question answers.

| | Model Family | # Parameters | # Examples | Accuracy (No CoT) | Accuracy (CoT) | Unfaithfulness |
|---|---|---|---|---|---|---|
| **AQuA-RAT** | FLAN-T5 | 77m | 254 | 23.23 | 22.83 | 46.06 |
| | FLAN-T5 | 248m | 254 | 25.20 | 19.29 | 31.89 |
| | FLAN-T5 | 783m | 254 | 24.02 | 23.62 | 28.35 |
| | FLAN-T5 | 2.85b | 254 | 24.02 | 25.59 | 21.65 |
| | FLAN-T5 | 11.3b | 254 | 27.56 | 22.44 | 30.31 |
| | FLAN-T5 | 20b | 254 | 27.56 | 29.13 | 29.13 |
| | Llama 2 | 7b | 254 | 22.44 | 22.44 | 45.67 |
| | Llama 2 | 13b | 254 | 20.87 | 24.41 | 40.16 |
| | Llama 2 | 70b | 254 | 25.98 | 35.43 | 39.37 |
| | Pythia DPO | 70m | 254 | 24.80 | 24.80 | 99.61 |
| | Pythia DPO | 160m | 254 | 24.80 | 21.65 | 64.57 |
| | Pythia DPO | 410m | 254 | 24.41 | 25.59 | 79.13 |
| | Pythia DPO | 1b | 254 | 20.47 | 18.50 | 6.69 |
| | Pythia DPO | 1.4b | 254 | 14.17 | 22.83 | 19.69 |
| | Pythia DPO | 2.8b | 254 | 23.23 | 22.05 | 25.59 |
| **ARC-Challenge** | FLAN-T5 | 77m | 1172 | 25.85 | 26.28 | 64.42 |
| | FLAN-T5 | 248m | 1172 | 38.74 | 38.57 | 62.20 |
| | FLAN-T5 | 783m | 1172 | 42.32 | 46.76 | 51.19 |
| | FLAN-T5 | 2.85b | 1172 | 64.85 | 60.15 | 60.58 |
| | FLAN-T5 | 11.3b | 1172 | 25.85 | 26.28 | 64.42 |
| | FLAN-T5 | 20b | 500 | 79.80 | 66.80 | 71.60 |
| | Llama 2 | 7b | 1172 | 57.59 | 58.36 | 66.64 |
| | Llama 2 | 13b | 697 | 63.41 | 67.00 | 70.88 |
| | Llama 2 | 70b | 988 | 76.72 | 76.62 | 76.42 |
| | Pythia DPO | 70m | 500 | 23.60 | 24.00 | 99.60 |
| | Pythia DPO | 160m | 1172 | 22.70 | 23.98 | 82.34 |
| | Pythia DPO | 410m | 1172 | 23.21 | 22.87 | 89.33 |
| | Pythia DPO | 1b | 500 | 21.80 | 28.00 | 35.60 |
| | Pythia DPO | 1.4b | 1172 | 26.28 | 27.13 | 16.47 |
| | Pythia DPO | 2.8b | 500 | 23.80 | 25.80 | 33.60 |
| **ARC-Easy** | FLAN-T5 | 77m | 2376 | 28.41 | 25.84 | 62.04 |
| | FLAN-T5 | 248m | 2376 | 53.20 | 51.18 | 65.57 |
| | FLAN-T5 | 783m | 2376 | 53.49 | 57.53 | 54.17 |
| | FLAN-T5 | 2.85b | 2376 | 28.41 | 25.84 | 62.04 |
| | FLAN-T5 | 11.3b | 2376 | 28.41 | 25.84 | 62.04 |
| | FLAN-T5 | 20b | 500 | 93.60 | 84.20 | 85.40 |
| | Llama 2 | 7b | 2376 | 75.42 | 72.77 | 75.93 |
| | Llama 2 | 13b | 1250 | 83.52 | 83.12 | 82.88 |
| | Llama 2 | 70b | 1098 | 92.35 | 91.80 | 91.89 |
| | Pythia DPO | 70m | 500 | 22.80 | 22.80 | 100.00 |
| | Pythia DPO | 160m | 2376 | 25.17 | 25.34 | 82.53 |
| | Pythia DPO | 410m | 2122 | 25.12 | 25.45 | 90.86 |
| | Pythia DPO | 1b | 500 | 23.20 | 26.60 | 39.60 |
| | Pythia DPO | 1.4b | 500 | 27.60 | 28.60 | 31.80 |
| | Pythia DPO | 2.8b | 500 | 24.40 | 24.40 | 34.20 |

Table 3: Full results for models evaluated on the multiple-choice benchmarks using the **original** answer choice ordering (continued in Tables 4-5).

| | Model Family | # Parameters | # Examples | Accuracy (No CoT) | Accuracy (CoT) | Unfaithfulness |
|---|---|---|---|---|---|---|
| **HellaSwag** | FLAN-T5 | 77m | 500 | 42.00 | 31.40 | 42.60 |
| | FLAN-T5 | 248m | 10042 | 42.70 | 38.41 | 59.04 |
| | FLAN-T5 | 783m | 10042 | 78.26 | 64.66 | 70.58 |
| | FLAN-T5 | 2.85b | 10042 | 91.04 | 86.21 | 89.73 |
| | FLAN-T5 | 11.3b | 10042 | 90.54 | 85.18 | 87.79 |
| | FLAN-T5 | 20b | 500 | 83.60 | 75.20 | 81.80 |
| | Llama 2 | 7b | 4434 | 51.71 | 44.72 | 55.28 |
| | Llama 2 | 13b | 627 | 59.49 | 53.43 | 60.29 |
| | Llama 2 | 70b | 500 | 72.01 | 62.69 | 69.03 |
| | Pythia DPO | 70m | 6891 | 25.13 | 25.03 | 98.77 |
| | Pythia DPO | 160m | 3940 | 24.97 | 25.00 | 99.52 |
| | Pythia DPO | 410m | 2117 | 25.22 | 25.22 | 96.03 |
| | Pythia DPO | 1b | 2731 | 25.38 | 24.79 | 34.60 |
| | Pythia DPO | 1.4b | 2236 | 24.82 | 23.61 | 32.07 |
| | Pythia DPO | 2.8b | 1536 | 25.52 | 27.08 | 24.54 |
| **LogiQA** | FLAN-T5 | 77m | 651 | 28.26 | 26.42 | 83.26 |
| | FLAN-T5 | 248m | 651 | 30.88 | 31.64 | 58.22 |
| | FLAN-T5 | 783m | 651 | 28.88 | 27.19 | 54.84 |
| | FLAN-T5 | 2.85b | 651 | 31.80 | 33.49 | 50.84 |
| | FLAN-T5 | 11.3b | 651 | 37.17 | 31.95 | 41.01 |
| | FLAN-T5 | 20b | 500 | 33.80 | 31.20 | 48.40 |
| | Llama 2 | 7b | 651 | 36.41 | 31.49 | 51.77 |
| | Llama 2 | 13b | 651 | 39.78 | 34.41 | 51.77 |
| | Llama 2 | 70b | 651 | 43.93 | 41.63 | 57.91 |
| | Pythia DPO | 70m | 500 | 20.40 | 20.20 | 99.40 |
| | Pythia DPO | 160m | 651 | 20.12 | 22.89 | 77.73 |
| | Pythia DPO | 410m | 651 | 20.74 | 21.81 | 90.48 |
| | Pythia DPO | 1b | 651 | 26.42 | 23.81 | 8.91 |
| | Pythia DPO | 1.4b | 500 | 24.80 | 22.60 | 25.00 |
| | Pythia DPO | 2.8b | 500 | 29.80 | 27.80 | 33.20 |
| **MMLU** | FLAN-T5 | 77m | 13985 | 27.93 | 26.05 | 38.30 |
| | FLAN-T5 | 248m | 13985 | 33.54 | 32.53 | 56.40 |
| | FLAN-T5 | 783m | 6379 | 42.19 | 35.57 | 44.19 |
| | FLAN-T5 | 2.85b | 500 | 42.20 | 43.40 | 46.60 |
| | FLAN-T5 | 11.3b | 500 | 48.00 | 37.20 | 40.40 |
| | FLAN-T5 | 20b | 500 | 50.40 | 46.40 | 50.20 |
| | Llama 2 | 7b | 4766 | 50.67 | 52.45 | 59.04 |
| | Llama 2 | 13b | 3520 | 53.81 | 55.62 | 63.07 |
| | Llama 2 | 70b | 500 | 54.80 | 64.40 | 56.60 |
| | Pythia DPO | 70m | 500 | 20.80 | 21.00 | 99.00 |
| | Pythia DPO | 160m | 2213 | 22.77 | 22.86 | 99.50 |
| | Pythia DPO | 410m | 1991 | 22.70 | 23.00 | 89.00 |
| | Pythia DPO | 1b | 500 | 24.21 | 21.43 | 35.32 |
| | Pythia DPO | 1.4b | 2172 | 27.85 | 27.03 | 33.38 |
| | Pythia DPO | 2.8b | 1533 | 27.27 | 25.83 | 33.66 |

Table 4: Full results for models evaluated on the multiple-choice benchmarks using the **original** answer choice ordering (continued in Table 5).

| | Model Family | # Parameters | # Examples | Accuracy (No CoT) | Accuracy (CoT) | Unfaithfulness |
|---|---|---|---|---|---|---|
| | FLAN-T5 | 77m | 500 | 27.80 | 21.60 | 59.60 |
| | FLAN-T5 | 248m | 500 | 46.00 | 41.40 | 67.20 |
| | FLAN-T5 | 783m | 500 | 44.80 | 47.60 | 47.20 |
| | FLAN-T5 | 2.85b | 500 | 70.40 | 65.20 | 64.00 |
| | FLAN-T5 | 11.3b | 500 | 74.40 | 33.40 | 29.20 |
| | FLAN-T5 | 20b | 500 | 79.60 | 69.60 | 71.20 |
| OpenBookQA | Llama 2 | 7b | 500 | 59.40 | 57.20 | 62.00 |
| | Llama 2 | 13b | 500 | 63.80 | 65.00 | 67.20 |
| | Llama 2 | 70b | 500 | 75.20 | 74.20 | 73.80 |
| | Pythia DPO | 70m | 500 | 27.60 | 28.20 | 89.60 |
| | Pythia DPO | 160m | 500 | 27.60 | 25.40 | 80.80 |
| | Pythia DPO | 410m | 500 | 27.60 | 25.80 | 91.80 |
| | Pythia DPO | 1b | 500 | 25.20 | 27.80 | 15.00 |
| | Pythia DPO | 1.4b | 500 | 26.80 | 23.80 | 34.80 |
| | Pythia DPO | 2.8b | 500 | 22.87 | 22.59 | 23.69 |
| | FLAN-T5 | 77m | 817 | 5.88 | 15.42 | 70.13 |
| | FLAN-T5 | 248m | 817 | 27.66 | 12.12 | 59.00 |
| | FLAN-T5 | 783m | 817 | 8.08 | 12.36 | 49.82 |
| | FLAN-T5 | 2.85b | 817 | 8.32 | 20.93 | 44.55 |
| | FLAN-T5 | 11.3b | 817 | 23.62 | 29.62 | 43.33 |
| | FLAN-T5 | 20b | 500 | 47.60 | 35.40 | 51.60 |
| TruthfulQA | Llama 2 | 7b | 817 | 28.89 | 35.37 | 58.02 |
| | Llama 2 | 13b | 817 | 24.11 | 46.39 | 45.41 |
| | Llama 2 | 70b | 817 | 41.98 | 45.78 | 57.04 |
| | Pythia DPO | 70m | 500 | 100.00 | 98.20 | 98.20 |
| | Pythia DPO | 160m | 817 | 99.88 | 66.95 | 66.83 |
| | Pythia DPO | 410m | 817 | 97.55 | 82.50 | 80.66 |
| | Pythia DPO | 1b | 817 | 10.16 | 60.59 | 13.71 |
| | Pythia DPO | 1.4b | 500 | 0.80 | 15.00 | 39.00 |
| | Pythia DPO | 2.8b | 500 | 10.80 | 22.60 | 31.20 |

Table 5: Full results for models evaluated on the multiple-choice benchmarks using the **original** answer choice ordering (continuation of Table 4).

| | Model Family | # Parameters | # Examples | Accuracy (No CoT) | Accuracy (CoT) | Unfaithfulness |
|---|---|---|---|---|---|---|
| **AQuA-RAT** | FLAN-T5 | 77m | 254 | 20.08 | 21.65 | 46.46 |
| | FLAN-T5 | 248m | 254 | 21.65 | 17.72 | 34.65 |
| | FLAN-T5 | 783m | 254 | 22.83 | 23.23 | 27.56 |
| | FLAN-T5 | 2.85b | 254 | 21.26 | 23.23 | 22.44 |
| | FLAN-T5 | 11.3b | 254 | 22.05 | 22.44 | 28.74 |
| | FLAN-T5 | 20b | 254 | 26.38 | 24.02 | 26.38 |
| | Llama 2 | 7b | 254 | 29.92 | 27.56 | 40.16 |
| | Llama 2 | 13b | 254 | 26.77 | 28.35 | 45.28 |
| | Llama 2 | 70b | 254 | 27.56 | 37.80 | 36.61 |
| | Pythia DPO | 70m | 254 | 19.69 | 20.08 | 99.61 |
| | Pythia DPO | 160m | 254 | 24.41 | 23.62 | 99.21 |
| | Pythia DPO | 410m | 254 | 18.91 | 18.57 | 85.43 |
| | Pythia DPO | 1b | 254 | 25.59 | 19.69 | 42.13 |
| | Pythia DPO | 1.4b | 254 | 20.87 | 18.54 | 20.47 |
| | Pythia DPO | 2.8b | 254 | 20.47 | 22.83 | 20.08 |
| **ARC-Challenge** | FLAN-T5 | 77m | 1172 | 24.23 | 22.27 | 65.10 |
| | FLAN-T5 | 248m | 1172 | 37.12 | 37.46 | 62.20 |
| | FLAN-T5 | 783m | 1172 | 43.00 | 46.42 | 49.91 |
| | FLAN-T5 | 2.85b | 1172 | 63.91 | 61.69 | 62.29 |
| | FLAN-T5 | 11.3b | 1172 | 25.43 | 25.60 | 62.20 |
| | FLAN-T5 | 20b | 500 | 78.80 | 65.80 | 69.40 |
| | Llama 2 | 7b | 1172 | 56.57 | 56.14 | 65.10 |
| | Llama 2 | 13b | 1172 | 62.37 | 63.14 | 68.60 |
| | Llama 2 | 70b | 1027 | 78.29 | 74.68 | 76.83 |
| | Pythia DPO | 70m | 500 | 24.60 | 24.80 | 99.80 |
| | Pythia DPO | 160m | 500 | 24.85 | 25.44 | 99.41 |
| | Pythia DPO | 410m | 500 | 25.00 | 25.40 | 84.00 |
| | Pythia DPO | 1b | 500 | 29.20 | 25.40 | 38.20 |
| | Pythia DPO | 1.4b | 389 | 24.42 | 22.11 | 32.39 |
| | Pythia DPO | 2.8b | 500 | 27.60 | 25.40 | 32.20 |
| **ARC-Easy** | FLAN-T5 | 77m | 2376 | 26.89 | 25.97 | 62.75 |
| | FLAN-T5 | 248m | 2376 | 53.83 | 49.16 | 65.24 |
| | FLAN-T5 | 783m | 2376 | 53.20 | 57.70 | 55.68 |
| | FLAN-T5 | 2.85b | 2376 | 27.48 | 26.85 | 61.24 |
| | FLAN-T5 | 11.3b | 2376 | 28.62 | 27.90 | 62.16 |
| | FLAN-T5 | 20b | 500 | 93.20 | 84.80 | 85.00 |
| | Llama 2 | 7b | 2376 | 75.97 | 72.35 | 75.51 |
| | Llama 2 | 13b | 1010 | 84.55 | 83.86 | 83.66 |
| | Llama 2 | 70b | 1111 | 93.07 | 90.91 | 91.09 |
| | Pythia DPO | 70m | 500 | 25.60 | 25.60 | 99.43 |
| | Pythia DPO | 160m | 500 | 28.60 | 28.60 | 98.87 |
| | Pythia DPO | 410m | 500 | 24.20 | 24.80 | 83.84 |
| | Pythia DPO | 1b | 500 | 27.60 | 23.40 | 33.60 |
| | Pythia DPO | 1.4b | 500 | 26.00 | 24.20 | 30.64 |
| | Pythia DPO | 2.8b | 500 | 28.60 | 26.40 | 31.43 |

Table 6: Full results for models evaluated on the multiple-choice benchmarks using the **same** answer choice ordering (continued in Tables 7-8).

| | Model Family | # Parameters | # Examples | Accuracy (No CoT) | Accuracy (CoT) | Unfaithfulness |
|---|---|---|---|---|---|---|
| **HellaSwag** | FLAN-T5 | 77m | 500 | 44.40 | 33.80 | 40.80 |
| | FLAN-T5 | 248m | 10042 | 42.52 | 38.24 | 59.91 |
| | FLAN-T5 | 783m | 10042 | 78.36 | 65.02 | 70.92 |
| | FLAN-T5 | 2.85b | 10042 | 90.83 | 86.32 | 89.65 |
| | FLAN-T5 | 11.3b | 10042 | 90.75 | 85.11 | 87.93 |
| | FLAN-T5 | 20b | 500 | 83.20 | 72.60 | 79.20 |
| | Llama 2 | 7b | 4557 | 50.34 | 43.16 | 55.48 |
| | Llama 2 | 13b | 3847 | 57.68 | 47.78 | 54.02 |
| | Llama 2 | 70b | 500 | 73.80 | 57.40 | 66.00 |
| | Pythia DPO | 70m | 500 | 27.60 | 27.40 | 98.20 |
| | Pythia DPO | 160m | 500 | 24.60 | 24.40 | 99.60 |
| | Pythia DPO | 410m | 500 | 24.63 | 24.63 | 91.71 |
| | Pythia DPO | 1b | 500 | 23.00 | 25.80 | 31.00 |
| | Pythia DPO | 1.4b | 500 | 24.20 | 26.20 | 37.00 |
| | Pythia DPO | 2.8b | 500 | 25.40 | 21.80 | 22.00 |
| **LogiQA** | FLAN-T5 | 77m | 651 | 24.58 | 24.58 | 84.95 |
| | FLAN-T5 | 248m | 651 | 29.49 | 27.50 | 56.53 |
| | FLAN-T5 | 783m | 651 | 24.27 | 27.65 | 56.22 |
| | FLAN-T5 | 2.85b | 651 | 32.72 | 32.72 | 48.39 |
| | FLAN-T5 | 11.3b | 651 | 36.56 | 31.03 | 39.78 |
| | FLAN-T5 | 20b | 500 | 37.00 | 33.80 | 48.40 |
| | Llama 2 | 7b | 651 | 34.41 | 33.64 | 51.31 |
| | Llama 2 | 13b | 651 | 36.87 | 33.33 | 51.31 |
| | Llama 2 | 70b | 651 | 42.09 | 39.63 | 58.53 |
| | Pythia DPO | 70m | 500 | 26.20 | 26.20 | 99.80 |
| | Pythia DPO | 160m | 500 | 25.40 | 25.40 | 99.80 |
| | Pythia DPO | 410m | 500 | 28.60 | 29.20 | 78.80 |
| | Pythia DPO | 1b | 500 | 19.70 | 25.12 | 33.00 |
| | Pythia DPO | 1.4b | 500 | 24.00 | 23.40 | 26.40 |
| | Pythia DPO | 2.8b | 500 | 25.00 | 23.40 | 33.00 |
| **MMLU** | FLAN-T5 | 77m | 13985 | 27.87 | 26.24 | 38.85 |
| | FLAN-T5 | 248m | 13985 | 33.96 | 33.26 | 54.67 |
| | FLAN-T5 | 783m | 6497 | 39.97 | 35.31 | 45.37 |
| | FLAN-T5 | 2.85b | 500 | 39.80 | 40.00 | 44.20 |
| | FLAN-T5 | 11.3b | 500 | 49.80 | 34.00 | 41.40 |
| | FLAN-T5 | 20b | 500 | 47.60 | 44.80 | 56.80 |
| | Llama 2 | 7b | 4901 | 50.17 | 49.89 | 59.36 |
| | Llama 2 | 13b | 3760 | 54.81 | 55.59 | 63.01 |
| | Llama 2 | 70b | 500 | 52.80 | 64.40 | 56.20 |
| | Pythia DPO | 70m | 500 | 29.00 | 29.00 | 98.80 |
| | Pythia DPO | 160m | 3904 | 26.46 | 26.43 | 99.46 |
| | Pythia DPO | 410m | 500 | 23.40 | 25.00 | 84.40 |
| | Pythia DPO | 1b | 2627 | 25.31 | 25.43 | 31.56 |
| | Pythia DPO | 1.4b | 2305 | 25.03 | 24.21 | 33.84 |
| | Pythia DPO | 2.8b | 1504 | 26.66 | 26.33 | 34.11 |

Table 7: Full results for models evaluated on the multiple-choice benchmarks using the **same** answer choice ordering (continued in Table 8).

| | Model Family | # Parameters | # Examples | Accuracy (No CoT) | Accuracy (CoT) | Unfaithfulness |
|---|---|---|---|---|---|---|
| **OpenBookQA** | FLAN-T5 | 77m | 500 | 24.40 | 25.40 | 61.40 |
| | FLAN-T5 | 248m | 500 | 28.60 | 24.40 | 61.60 |
| | FLAN-T5 | 783m | 500 | 27.20 | 28.20 | 61.80 |
| | FLAN-T5 | 2.85b | 500 | 29.60 | 27.00 | 58.80 |
| | FLAN-T5 | 11.3b | 500 | 26.20 | 24.00 | 59.80 |
| | FLAN-T5 | 20b | 500 | 79.20 | 70.40 | 72.80 |
| | Llama 2 | 7b | 500 | 58.40 | 55.80 | 61.20 |
| | Llama 2 | 13b | 500 | 65.00 | 63.80 | 65.60 |
| | Llama 2 | 70b | 500 | 80.00 | 73.40 | 74.20 |
| | Pythia DPO | 70m | 500 | 24.60 | 25.20 | 99.40 |
| | Pythia DPO | 160m | 500 | 24.20 | 24.20 | 99.80 |
| | Pythia DPO | 410m | 500 | 25.60 | 26.60 | 90.80 |
| | Pythia DPO | 1b | 500 | 24.00 | 22.60 | 34.80 |
| | Pythia DPO | 1.4b | 500 | 25.80 | 26.40 | 32.80 |
| | Pythia DPO | 2.8b | 500 | 24.74 | 26.48 | 26.83 |
| **TruthfulQA** | FLAN-T5 | 77m | 817 | 19.34 | 20.81 | 71.73 |
| | FLAN-T5 | 248m | 817 | 18.97 | 21.79 | 71.24 |
| | FLAN-T5 | 783m | 817 | 23.26 | 23.87 | 70.62 |
| | FLAN-T5 | 2.85b | 817 | 18.85 | 19.83 | 70.75 |
| | FLAN-T5 | 11.3b | 817 | 20.93 | 20.93 | 71.60 |
| | FLAN-T5 | 20b | 500 | 46.20 | 36.60 | 55.80 |
| | Llama 2 | 7b | 817 | 28.15 | 43.21 | 53.98 |
| | Llama 2 | 13b | 817 | 31.09 | 47.86 | 50.92 |
| | Llama 2 | 70b | 809 | 46.35 | 56.00 | 60.44 |
| | Pythia DPO | 70m | 500 | 26.00 | 26.60 | 99.20 |
| | Pythia DPO | 160m | 500 | 21.40 | 21.60 | 99.20 |
| | Pythia DPO | 410m | 500 | 24.00 | 22.80 | 82.20 |
| | Pythia DPO | 1b | 500 | 23.60 | 21.20 | 40.80 |
| | Pythia DPO | 1.4b | 500 | 24.40 | 23.40 | 45.80 |
| | Pythia DPO | 2.8b | 500 | 23.40 | 24.80 | 34.40 |

Table 8: Full results for models evaluated on the multiple-choice benchmarks using the **same** answer choice ordering (continuation of Table 7).

| | Model Family | # Parameters | # Examples | Accuracy (No CoT) | Accuracy (CoT) | Unfaithfulness |
|---|---|---|---|---|---|---|
| **AQuA-RAT** | FLAN-T5 | 77m | 254 | 24.02 | 20.08 | 22.44 |
| | FLAN-T5 | 248m | 254 | 15.75 | 23.62 | 23.23 |
| | FLAN-T5 | 783m | 254 | 22.44 | 23.23 | 25.20 |
| | FLAN-T5 | 2.85b | 254 | 22.44 | 28.74 | 25.59 |
| | FLAN-T5 | 11.3b | 254 | 24.02 | 27.56 | 21.26 |
| | FLAN-T5 | 20b | 254 | 27.56 | 29.53 | 31.10 |
| | Llama 2 | 7b | 254 | 24.41 | 24.41 | 22.83 |
| | Llama 2 | 13b | 254 | 22.83 | 25.59 | 26.77 |
| | Llama 2 | 70b | 254 | 22.83 | 38.19 | 24.41 |
| | Pythia DPO | 70m | 254 | 18.90 | 19.29 | 24.02 |
| | Pythia DPO | 160m | 254 | 16.54 | 19.29 | 16.14 |
| | Pythia DPO | 410m | 254 | 21.26 | 25.59 | 21.26 |
| | Pythia DPO | 1b | 254 | 21.65 | 21.65 | 18.90 |
| | Pythia DPO | 1.4b | 254 | 20.08 | 20.08 | 20.87 |
| | Pythia DPO | 2.8b | 254 | 22.83 | 20.87 | 24.41 |
| **ARC-Challenge** | FLAN-T5 | 77m | 1172 | 24.15 | 25.60 | 22.95 |
| | FLAN-T5 | 248m | 1172 | 38.82 | 36.60 | 52.30 |
| | FLAN-T5 | 783m | 1172 | 43.43 | 42.75 | 40.70 |
| | FLAN-T5 | 2.85b | 1172 | 64.33 | 59.73 | 59.73 |
| | FLAN-T5 | 11.3b | 1172 | 26.11 | 26.19 | 26.19 |
| | FLAN-T5 | 20b | 500 | 78.60 | 66.00 | 67.60 |
| | Llama 2 | 7b | 1172 | 55.20 | 54.44 | 52.13 |
| | Llama 2 | 13b | 1172 | 64.16 | 65.44 | 62.54 |
| | Llama 2 | 70b | 988 | 77.13 | 77.02 | 75.20 |
| | Pythia DPO | 70m | 500 | 25.00 | 26.20 | 23.00 |
| | Pythia DPO | 160m | 500 | 25.42 | 25.06 | 22.81 |
| | Pythia DPO | 410m | 500 | 26.00 | 23.40 | 24.60 |
| | Pythia DPO | 1b | 500 | 25.20 | 25.40 | 22.80 |
| | Pythia DPO | 1.4b | 500 | 22.81 | 26.32 | 25.00 |
| | Pythia DPO | 2.8b | 500 | 24.60 | 28.00 | 22.00 |
| **ARC-Easy** | FLAN-T5 | 77m | 2376 | 27.06 | 25.59 | 25.25 |
| | FLAN-T5 | 248m | 2376 | 53.07 | 49.16 | 58.96 |
| | FLAN-T5 | 783m | 2376 | 53.41 | 57.45 | 46.59 |
| | FLAN-T5 | 2.85b | 2376 | 24.49 | 25.34 | 25.84 |
| | FLAN-T5 | 11.3b | 2376 | 26.77 | 27.61 | 24.49 |
| | FLAN-T5 | 20b | 500 | 93.60 | 83.40 | 83.60 |
| | Llama 2 | 7b | 2376 | 74.71 | 71.04 | 67.26 |
| | Llama 2 | 13b | 2376 | 81.57 | 79.04 | 75.38 |
| | Llama 2 | 70b | 1109 | 91.61 | 91.07 | 89.18 |
| | Pythia DPO | 70m | 500 | 25.40 | 26.40 | 25.00 |
| | Pythia DPO | 160m | 500 | 22.87 | 26.52 | 22.63 |
| | Pythia DPO | 410m | 500 | 24.70 | 29.17 | 26.49 |
| | Pythia DPO | 1b | 500 | 26.00 | 23.20 | 23.00 |
| | Pythia DPO | 1.4b | 500 | 25.40 | 25.20 | 27.00 |
| | Pythia DPO | 2.8b | 500 | 26.80 | 24.40 | 24.20 |

Table 9: Full results for models evaluated on the multiple-choice benchmarks using the **different** answer choice ordering (continued in Tables 10-11).

| | Model Family | # Parameters | # Examples | Accuracy (No CoT) | Accuracy (CoT) | Unfaithfulness |
|---|---|---|---|---|---|---|
| HellaSwag | FLAN-T5 | 77m | 500 | 45.99 | 28.47 | 35.77 |
| | FLAN-T5 | 248m | 10042 | 42.57 | 38.56 | 55.28 |
| | FLAN-T5 | 783m | 10042 | 77.90 | 64.51 | 68.00 |
| | FLAN-T5 | 2.85b | 10042 | 90.64 | 85.61 | 88.22 |
| | FLAN-T5 | 11.3b | 10042 | 90.38 | 85.39 | 86.09 |
| | FLAN-T5 | 20b | 500 | 82.80 | 71.80 | 75.60 |
| | Llama 2 | 7b | 4358 | 51.68 | 44.63 | 48.81 |
| | Llama 2 | 13b | 1975 | 58.68 | 51.44 | 54.84 |
| | Llama 2 | 70b | 588 | 73.13 | 58.16 | 66.67 |
| | Pythia DPO | 70m | 500 | 22.20 | 26.00 | 25.60 |
| | Pythia DPO | 160m | 500 | 23.00 | 23.40 | 23.80 |
| | Pythia DPO | 410m | 500 | 25.40 | 24.80 | 23.80 |
| | Pythia DPO | 1b | 500 | 20.53 | 26.24 | 25.10 |
| | Pythia DPO | 1.4b | 500 | 24.20 | 24.00 | 27.20 |
| | Pythia DPO | 2.8b | 500 | 25.68 | 27.05 | 28.42 |
| LogiQA | FLAN-T5 | 77m | 651 | 24.12 | 24.73 | 27.96 |
| | FLAN-T5 | 248m | 651 | 28.26 | 29.03 | 48.69 |
| | FLAN-T5 | 783m | 651 | 29.34 | 27.65 | 33.64 |
| | FLAN-T5 | 2.85b | 651 | 27.65 | 32.10 | 42.55 |
| | FLAN-T5 | 11.3b | 651 | 37.33 | 29.65 | 38.40 |
| | FLAN-T5 | 20b | 500 | 38.60 | 34.20 | 45.40 |
| | Llama 2 | 7b | 651 | 34.25 | 32.26 | 44.55 |
| | Llama 2 | 13b | 651 | 35.48 | 36.71 | 43.47 |
| | Llama 2 | 70b | 651 | 42.86 | 38.86 | 52.69 |
| | Pythia DPO | 70m | 500 | 23.80 | 24.80 | 24.40 |
| | Pythia DPO | 160m | 500 | 23.20 | 23.20 | 25.80 |
| | Pythia DPO | 410m | 500 | 25.80 | 22.40 | 29.20 |
| | Pythia DPO | 1b | 500 | 23.60 | 23.80 | 26.80 |
| | Pythia DPO | 1.4b | 500 | 24.20 | 25.40 | 27.00 |
| | Pythia DPO | 2.8b | 500 | 23.80 | 21.40 | 28.20 |
| MMLU | FLAN-T5 | 77m | 13985 | 28.24 | 26.04 | 30.22 |
| | FLAN-T5 | 248m | 13985 | 33.66 | 33.11 | 50.45 |
| | FLAN-T5 | 783m | 6158 | 41.51 | 35.37 | 43.54 |
| | FLAN-T5 | 2.85b | 500 | 44.20 | 37.80 | 46.00 |
| | FLAN-T5 | 11.3b | 500 | 48.80 | 35.40 | 37.40 |
| | FLAN-T5 | 20b | 500 | 45.20 | 48.00 | 52.20 |
| | Llama 2 | 7b | 4698 | 49.98 | 49.17 | 51.34 |
| | Llama 2 | 13b | 3628 | 55.40 | 56.89 | 56.01 |
| | Llama 2 | 70b | 500 | 56.00 | 63.00 | 53.80 |
| | Pythia DPO | 70m | 500 | 24.00 | 24.60 | 23.80 |
| | Pythia DPO | 160m | 3718 | 25.26 | 24.93 | 25.34 |
| | Pythia DPO | 410m | 500 | 24.00 | 24.00 | 24.20 |
| | Pythia DPO | 1b | 2565 | 25.54 | 23.90 | 25.07 |
| | Pythia DPO | 1.4b | 2224 | 25.40 | 26.17 | 25.58 |
| | Pythia DPO | 2.8b | 500 | 25.60 | 22.80 | 21.00 |

Table 10: Full results for models evaluated on the multiple-choice benchmarks using the **different** answer choice ordering (continued in Table 11).

| | Model Family | # Parameters | # Examples | Accuracy (No CoT) | Accuracy (CoT) | Unfaithfulness |
|---|---|---|---|---|---|---|
| **OpenBookQA** | FLAN-T5 | 77m | 500 | 26.40 | 24.20 | 28.60 |
| | FLAN-T5 | 248m | 500 | 23.80 | 25.60 | 28.40 |
| | FLAN-T5 | 783m | 500 | 24.60 | 28.40 | 27.60 |
| | FLAN-T5 | 2.85b | 500 | 27.80 | 25.00 | 28.20 |
| | FLAN-T5 | 11.3b | 500 | 31.40 | 25.60 | 27.00 |
| | FLAN-T5 | 20b | 500 | 78.40 | 69.20 | 68.80 |
| | Llama 2 | 7b | 500 | 57.60 | 57.60 | 57.80 |
| | Llama 2 | 13b | 500 | 63.40 | 63.80 | 57.00 |
| | Llama 2 | 70b | 500 | 77.60 | 76.40 | 73.00 |
| | Pythia DPO | 70m | 500 | 30.00 | 40.00 | 10.00 |
| | Pythia DPO | 160m | 500 | 23.35 | 21.83 | 21.83 |
| | Pythia DPO | 410m | 500 | 22.00 | 24.00 | 26.80 |
| | Pythia DPO | 1b | 500 | 24.40 | 25.80 | 21.60 |
| | Pythia DPO | 1.4b | 500 | 25.40 | 23.00 | 25.60 |
| | Pythia DPO | 2.8b | 500 | 24.00 | 25.80 | 25.00 |
| **TruthfulQA** | FLAN-T5 | 77m | 500 | 22.77 | 20.44 | 22.03 |
| | FLAN-T5 | 248m | 500 | 21.54 | 21.79 | 22.28 |
| | FLAN-T5 | 783m | 500 | 20.20 | 20.20 | 22.40 |
| | FLAN-T5 | 2.85b | 500 | 22.40 | 23.62 | 21.91 |
| | FLAN-T5 | 11.3b | 500 | 21.30 | 23.99 | 22.64 |
| | FLAN-T5 | 20b | 500 | 48.00 | 36.40 | 52.40 |
| | Llama 2 | 7b | 500 | 30.97 | 40.51 | 39.90 |
| | Llama 2 | 13b | 500 | 32.93 | 48.96 | 47.98 |
| | Llama 2 | 70b | 500 | 46.88 | 56.79 | 57.77 |
| | Pythia DPO | 70m | 500 | 23.80 | 25.00 | 25.60 |
| | Pythia DPO | 160m | 500 | 26.60 | 26.40 | 25.40 |
| | Pythia DPO | 410m | 500 | 26.80 | 23.00 | 23.00 |
| | Pythia DPO | 1b | 500 | 26.80 | 25.80 | 30.20 |
| | Pythia DPO | 1.4b | 500 | 22.40 | 23.20 | 27.20 |
| | Pythia DPO | 2.8b | 500 | 26.00 | 23.00 | 26.00 |

Table 11: Full results for models evaluated on the multiple-choice benchmarks using the **different** answer choice ordering (continuation of Table 10).

