# OpenReview forum: "Chain-of-Thought Unfaithfulness as Disguised Accuracy"
_TMLR — Accepted by TMLR_

### Review · Reviewer_u5Tm · 2024-03-12

**Summary Of Contributions:**

This paper reproduces part of the results of Lanham et al. on unfaithfulness in CoT generations for 3 open LLM families. "Faithfulness" measures the extent to which an explanation accurately represents the reasoning process behind a model prediction. Lanham et al, motivated by the benefit of using CoTs for interpretability if they are in fact faithful to a model's reasoning, define a metric to measure faithfulness. This metric counts how often a zero-shot prediction is the same as a CoT prediction, the more often the more unfaithful. This assumes that if a model doesn't need the CoT (i.e. can produce the same answer zero-shot), the CoT will be less faithful. Lanham et al. motivate this metric by showing it correlates with a more complex intervention-based metric; if one intervenes on the CoT and the answer changes, the model is more faithful (e.g. intervening by adding a mistake in the CoT, truncating it, paraphrasing it, or adding filler tokens). The submitted paper reproduces the results of Lanham et al's simpler unfaithfulness metric that counts how often an answer changes from zero-shot to CoT for three different LLM families of different sizes. The result is that similar scaling behaviour as Lanham et al report is observed for models that get non-trivial performance. The finding of Lanham et al that faithfulness is related to task difficulty is not observed in the authors experiments. The authors do an additional experiment where they order the multiple-choice questions differently between zero-shot and CoT prompts.

**Audience:**

Yes

**Broader Impact Concerns:**

N.A.

**Claims And Evidence:**

Yes

**Requested Changes:**

**Necessary**: The results as an average over benmchmarks are really difficult, if not impossible, to interpret like this. The variance between benchmarks seems like a weird thing to report and misleading; to me looking at these plots I first assume variance is too high to make any conclusions. It would be more useful to also report whether it's inverse scaling for each benchmark separately, or only for a few representative benchmarks. If you show the average, I wouldn't show variance over benchmarks. You still have space so I would make an additional plot that breaks down more per benchmark.

**Necessary**: address the issue with the confounder factor introduced by having a different ordering between zero-shot and CoT prompt by reporting how often the model changes it's answer for the same prompt but different ordering of the answer; if this is high, the "different" condition cannot say much about unfaithfulness.

**Necessary**: also reproduce the intervention experiments done by Lanham et al. on the open-access LLMs to perform that also for these open-access LLMs, the simpler unfaithfulness metric correlates with conclusions from intervention techniques.

**Minor but necessary**:  It's unclear what "as a property of all LLMs" means, this is too broad a claim. Would change to something about it generalizing to different, non-proprietary, LLM families, but not to "all LLMs".

**Would make the paper better** Only in figure 3 we see Pythia performs randomly for both zero-shot and CoT. The fact that the unfaithfulness metric doesn't work for random performing models is mentioned by you in the paper, but I believe it needs to be clearer in the text. I would propose an additional plot that first shows the accuracies of the models and only then faithfulness. Alternatively, also in figure 1, I think potentially overlaying accuracy on the same plot (with a right axis) would make sense.

**Comments on writing**
Abstract: This is personal preference so feel free to ignore, but I would slightly alter the order of information in the abstract such that the reader knows what the paper is about before having to read a few sentences about a related work. E.g. "Understanding the extent to which Chain-of-Thought (CoT) generations align with a large language model’s (LLM) internal computations is critical for deciding whether to trust an LLM’s output. In this work we replicate findings .."

Abstract: I think "V-shaped scaling behaviour" is clearer than "scaling-then-inverse-scaling relationship" as used in the abstract

**Strengths And Weaknesses:**

**Strengths**

- It's very important for science to reproduce results of experiments on proprietary models with open-access models.

- The authors reproduce results of the original paper on three different open-access model families.

**Weaknesses**

- Only part of the Lanham results are reproduced, which make the conclusions of the paper not sound. The original authors (Lanham et al.) claim this metric measures unfaithfulness **by virtue of it's correlation with another method that measures faithfulness**. In and of itself, the metric that is used in the submission need not say anything about faithfulness. If an answer changes from zero-shot to CoT, that does not need to indicate faithfulness. If the answer does not change, it might mean the model just also knows the answer zero-shot, and with CoT it still follows the reasoning from the CoT faithfully. The model might even employ a different reasoning for zero-shot than CoT, this metric says nothing about that. Further, if the answer does change from zero-shot to CoT, that also does not mean the model is faithful to it's reasoning. The authors are up front about these limitations here and there in the paper, but the problem is that the experiment that the original Lanham paper does (intervening on the inputs to the model), is not done. Without this experiment, and showing that indeed that correlates with the unfaithfulness metric for the open-access models, you cannot say something about how well this unfaithfulness metric works for other LLM families than presented in Lanham et al.

- The additional experiments the authors propose (ordering MC answers differently between zero-shot and CoT) are not sound due to a confounder that is unaddressed. The authors adjust the original unfaithfulness metric, which tests whether an answer changes if using CoT prompting instead of zero-shot prompting, by also changing the ordering of answers between zero-shot and CoT prompt. This is explicitly introducing a confounder, and experiments need to be done to test if this is the reason for the different unfaithfulness results due to this metric. The authors seemingly have the data necessary to test for this confounder already; does a models answer change within prompt if you change the ordering? I.e., if the answer changes from zero-shot to zero-shot different ordering, perhaps an answer change from zero-shot to CoT doesn't mean that the model is more faithful, but just that it's sensitive to answer orders.

- The above makes me wonder if one of the main conclusions of this paper holds: because you observe a high correlation in the different condition between accuracy and unfaithfulness you claim that unfaithfulness as proposed by Lanham et al is too reductive. I think the metric is indeed too reductive, but not because of the presented correlation. As mentioned above, I think the "different" condition introduces a confounder which makes it essentially a different and less sound metric than the one Lanham et al propose. You first need to investigate the confounder that the different condition introduces before you can say anything about the original unfaithfulness metric. Furthermore, even if the confounder is addressed, a high correlation between accuracy and unfaithfulness doesn't necessarily mean there is something wrong with the metric. It might really just be that models that have more capabilities are less faithful.

- The presentation of the results is very difficult to interpret because the authors present an average over about 8 different benchmarks (of which 1 is MMLU). An average like that is really difficult to interpret, and a variance over benchmarks seems even less useful. For example, it could now be that for certain benchmarks the unfaithfulness becomes less with scale.

---

> ### Author Response · Authors · 2024-05-14
> **Response to Reviewer u5Tm**
>
> We thank Reviewer u5Tm for the careful review of our paper! We appreciate that the reviewer finds our efforts to reproduce findings from proprietary models on open models to be important.
>
> ### **Re: The scaling of CoT faithfulness with unfaithfulness measurements based on early answering and adding mistakes.**
>
> Our research questions are focused on replicating a scaling-then-inverse-scaling relationship between model size and one of the measures of faithfulness by Lanham et al. These results are presented in Section 3 of their work where they say the following about the faithfulness measurement we adopt:
>
> > This metric is highly predictive of overall early answering and adding mistakes results […] We [Lanham et al.] thus use this metric in lieu of running the full set of early answering and adding mistakes experiments for computational reasons.
>
> That is, **Lanham et al. themselves exclusively use this measurement to analyze the scaling of CoT faithfulness**, and we follow their experimental setup as closely as possible. We stress this in a new Footnote 1.
>
> Furthermore, the suggestion to repeat our experiments using their other faithfulness measurements (based on early answering and adding mistakes) makes us wonder whether we should expect significant differences from these measurements given that they are highly correlated with the one used. We do not anticipate so, given the high correlation and given that the authors themselves deem these measurements interchangeable.
>
> ---
>
> ### **Re: Adjusting the unfaithfulness measurement to consider if a model’s answer changes within the same prompt if the answer choice order is changed.**
>
> We include the `diff` setting motivated by the known inductive bias in models, where they consistently select the same answer choice. Our aim is to demonstrate that this sensitivity to answer choice order can make the proposed unfaithfulness measurement appear high, not due to actual CoT unfaithfulness, but because of the order sensitivity. For example, low-accuracy models cannot answer the question with or without CoT, but they consistently pick the same answer choice, falsely indicating unfaithfulness according to Lanham et al.'s metric.
>
> However, **Reviewer u5Tm inspired us to make this point more clearly. In place of `diff`, we introduce a new measurement and revise the paper throughout**. Specifically, we normalize the unfaithfulness measurement in the `same` setting as follows:
>
> $$\text{Unfaithfulness}_{\text{Normalized}}(\text{CoT})=\frac{\text{percentage of questions for which the answer is the same with CoT and zero-shot}}{\text{percentage of questions for which the answer choice is the same with zero-shot when prompted twice with different answer choice orderings}}$$
>
>
> This measures the frequency of answer changes with CoT, compared to changes expected from merely shuffling the answer order. **Figure 2, middle, now includes the plot with this measurement. There is again a notable drop for the smaller, low-performing models. The strong correlation between accuracy and this unfaithfulness measurement is available in Figure 4 (in place of the previous one).** The measurement is defined in Section 3 and the `diff` results are available in Appendix A.2.
>
> ---
>
> ### **Re: Discussion on the possibility of CoTs becoming more unfaithful as models get larger.**
>
> Indeed, that could be the case, but there are valid reasons to question this strongly. In the Discussion section, we say:
> > If these CoTs are also regularly plausible — which is often the case with highly capable models like GPT-4 — the correlation suggests that such
> **models reason differently from people in all cases where they solve the task well.**
>
> ---
>
> ### **Re: “The presentation of the results is very difficult to interpret because the authors present an average over about 8 different benchmarks (of which 1 is MMLU).”**
>
> Figures 9 and 10 in the Appendix, along with corresponding Tables 3-11, show the results per benchmark, albeit with MMLU still as a single aggregated benchmark. Given that more detailed tables in the Appendix support our findings, we chose to present the data as an average over benchmarks to prevent overwhelming the reader with results. Aggregating MMLU is consistent with Lanham et al.’s approach.

---

> > ### Comment · Reviewer_GdPT · 2024-05-14
> > **I would be curious for Reviewer u5Tm's follow-up here**
> >
> > I do not fully understand the concern with the confound in the order flipping experiments, but it seems to me the main potential limitation in the paper, so I'd be curious to get Reviewer u5Tm's sense for the state of the things after the authors' updates. Do you think the concern was resolved?

---

> > ### Comment · Reviewer_u5Tm · 2024-05-15
> > **Response to author response about not reproducing intervention experiments**
> >
> > Though it's true that Lanham et al. exclusively use the metric you use for the scaling experiments, they justify this through the correlation they find. In your case, we do not know whether the metric correlates with intervention experiments for other families of non-proprietary LLMs. Now you're essentially using results from a paper you're aiming to reproduce to justify the approach you're taking. Let's say the intervention metric does not correlate with the zero-shot to CoT answer change metric for the models you test; then Lanham et al most likely wouldn't have used this simpler metric to do the scaling experiments either. The way I interpret the metric Lanham et al. propose is mainly the intervention-based metric, and only if it correlates with the simpler metric you can also use that one.
> >
> > In any case, given that the other reviewers don't share my concerns, and given that the conclusion of the current submission remains valid whether or not the metric correlates with the intervention experiments, I can soften the requirement in my original review a bit, and not ask for reproducing the intervention results. Instead I'd like the authors to caveat the results and add a discussion point. Be clearer about the fact that you don't reproduce all of their experiments and mention in a discussion that it could be that there's no selection bias in the smaller models Lanham et al. use but that this is impossible to test because they are proprietary. One suggested change would be in the abstract, replacing "we replicate their experimental setup" by "we replicate their experimental setup in the scaling experiments" or "we replicate part of their experimental setup". This might seem minor, but the abstract threw me off initially because I expected Lanham et al. to be fully reproduced.

---

> > > ### Author Response · Authors · 2024-05-15
> > > **Re: Response to author response about not reproducing intervention experiments**
> > >
> > > Thank you for clarifying! These are good points and we integrate them as follows.
> > >
> > > In the Abstract, we say:
> > >
> > > > We replicate the experimental setup in **their section focused on scaling experiments** with three different families of models and, under specific conditions, successfully reproduce the scaling trends for CoT faithfulness they report.
> > >
> > > In Experimental Setup:
> > >
> > > > In this section, we describe the metrics, models, tasks, and implementation details we followed **to reproduce the scaling experiments.**
> > >
> > > We bold phrases only in this comment to clarify where we stress that a portion of Lanham et al.’s results is replicated.
> > >
> > > In the Discussion, we add:
> > >
> > > > While it is possible that the smaller models used by Lanham et al. (2023) do not exhibit such a bias, which we cannot test directly since they are proprietary, its existence in the models we test highlights a potential flaw in the unnormalized metric.

---

> > ### Comment · Reviewer_u5Tm · 2024-05-15
> > **Response to author response on aggregating metrics over many benchmarks**
> >
> > I understand the choice and I have no issue with aggregating MMLU, but given that it's impossible to be sure about the main result in your paper (reproducing inverse scaling results) without consulting the appendix, I would still suggest adding the per-benchmark results in the main paper, e.g. figure 9, or similarly to how Lanham et al. present it, which is broken down per benchmark in every figure.

---

> > > ### Author Response · Authors · 2024-05-15
> > > **RE: Response to author response on aggregating metrics over many benchmarks**
> > >
> > > Makes sense. We revised Figure 2 accordingly.

---

> ### Comment · Reviewer_u5Tm · 2024-05-15
> **Response to Reviewer GdPT and the authors on my concern about the confounder introduced by the different ordering**
>
> Thanks to the authors for addressing this point and for flagging the response, reviewer GdPT. This particular concern has been addressed by the author response. To clarify what my concern was; if you change the prompt in two ways (different ordering and adding a CoT), and measure how often the model changes its answer, it could be both due to the different ordering (as LLMs are known to be sensitive to MC ordering (e.g. see Alzahrani. et al, 2024 and many citations in that work)) and due to the addition of the CoT, meaning you still don't know for sure if the model is sensitive to changes in answer order. However, the authors addressed this point by separately measuring the answer change due to addition of CoT, and the answer change by shuffling the order in the zero-shot setting. Now, the experiment more convincingly indicates to me that the smaller models are not necessarily more faithful, but might just be more sensitive to different MC ordering.

---

### Review · Reviewer_GdPT · 2024-03-17

**Summary Of Contributions:**

The paper presents a clean experiment which indicates that a chain-of-thought faithfulness metric from [Lanham et al. 2023](https://arxiv.org/abs/2307.13702) may be confounded.  Specifically, the average difference in answer between CoT vs. NoCoT depends wildly on whether it is measured with the same random answer ordering for both CoT and NoCoT or different answer ordering.  In particular, weak models may seem more faithful because they derive more of their answer distribution from letter preference (e.g., for A).

The paper is clean and to-the-point, and I believe it should be accepted.

**Audience:**

Yes

**Broader Impact Concerns:**

No ethical concerns: the paper shows a potential confound in a method for investigating safety concerns (whether humans would infer the correct things from seeing chain-of-thought transcripts), which is a good safety contribution.

**Claims And Evidence:**

Yes

**Requested Changes:**

The most important one is to determine what ordering Lanham et al. used, to make the comparisons easier.

An improvement to Figure 4 to make the x-axis smoother would be nice as well, though I realize that may be hard.  If no improvement is feasible, it may be worth caveating the results a bit.

**Strengths And Weaknesses:**

### Strengths

1. The main Figure 2 is terrific!  The top row showing wild differences in faithfulness metric is quite the contrast with the bottom row showing no difference is accuracy (as would be expected, so good control).  This is a convincing warning sign about the metric.
2. Figure 3 showing the accuracy/faithfulness correlation with different orderings is very nice as well, and I agree it's suspicious.
2. The papers conclusions and strength of conclusions are well calibrated to the experiments run: it's hard to be certain what it is going on, and the paper does not overclaim.  In particular, I found it tempting through out to go into more philosophical objections to the faithfulness metrics, but it is reasonable to keep this paper scoped to concrete empirical concerns.

### Weaknesses

1. We don't know what ordering method Lanham et al. used!  You should ask them!  I realize it's hard to do this with deanonymizing, but the paper would be significantly more useful if one could easily compare with the plots in Lanham et al.  If the authors don't want to not deanonymize, perhaps the TMLR editor could ask?
2. It's sad that the x-axis in Figure 4 is so sparse, and the conclusion (that the optimally faithful model is the same size across problems) seems weak as a result.  I would suggest either caveating this plot, or finding a different model family with a smoother range of sizes for the experiment.

---

> ### Author Response · Authors · 2024-05-14
> **Response to Reviewer GdPT**
>
> We thank reviewer GdPT for the time spent carefully reviewing our paper! We appreciate that the reviewer found our experiments and plots illustrating a confounder in Lanham et al.’s faithfulness metric to be convincing. We are hopeful that our additional experiment (requested by u5Tm) will make our argument even more clear.
>
> ### **Re: Which ordering Lanham et al. used?**
>
> We agree that directly comparing methods would be helpful for determining reproducibility; however, in our experiments we have covered all of the different possibilities of answer choice orderings (full results of these can be found in the appendix). Additionally, we include a new normalized version of the metric which directly accounts for the sensitivity to choice order. In the context of the unfaithfulness metric correlating with accuracy and dropping when the sensitivity to answer choice order is considered, we don’t believe knowing this implementation detail would change our argument about the reliability of the metric.
>
> ---
>
> ### **Re: The sparsity of the x-axis in Figure 4.**
>
> We agree that increasing the number of model sizes in Figure 3 (previously Figure 4) would further strengthen our argument; however, we’ve found that for this task, smaller models (e.g. Pythia) where we have more size options tend not to have meaningful performance. Lanham et al.’s arithmetic results exhibit similar sparsity on the basis that their smaller models (<2 billion parameters) struggle to produce arithmetic results altogether. We agree that this point needs to be clear to the reader. In the last paragraph of section 4, we write:
>
> > The discrepancy in our findings compared to Lanham et al. (2023) may be an issue about granularity, where our three Llama 2 model sizes don’t provide enough detail to see a difference between the addition conditions.”
>
> We have also added this caveat to the caption of Figure 3:
>
> > For both tasks the optimally faithful model according to the metric occurs at 13B; however, this might be due to the sparse nature of the x-axis.

---

### Review · Reviewer_H9RD · 2024-04-29

**Summary Of Contributions:**

This paper investigates the reliability of chain-of-thought (CoT) reasoning in large language models (LLMs). It explores how faithfully these models represent their internal reasoning processes when generating CoT outputs. The study aims to assess the extent to which CoT generations align with the internal computations of LLMs and to question the trustworthiness of LLM outputs based on CoT reasoning. It builds on prior work by Lanham et al. (2023) that proposed a metric for evaluating the faithfulness of CoT by examining if the model's answers change when CoT is applied compared to standard prompting. The authors replicate the methodology of Lanham et al. across multiple LLM families, including proprietary models of varying sizes (from 810 million to 175 billion parameters), to see if the previous findings generalize. They introduce a metric that quantifies unfaithfulness based on whether the model's output changes with the introduction or removal of CoT. The paper evaluates this across various NLP benchmarks and controlled task settings.

The paper reports a scaling-then-inverse-scaling relationship between model size and faithfulness, where mid-sized models (around 13 billion parameters) show higher faithfulness compared to both smaller and larger models. However, a critical finding is that simply changing the order of answer choices can drastically alter the faithfulness metric, reducing it by up to 73 percentage points. This suggests the metric may not robustly measure faithfulness. There is also a high correlation ($R^2 = 0.91$) between the faithfulness metric and task accuracy, which raises questions about whether the metric truly captures faithfulness or merely accuracy.

**Audience:**

Yes

**Claims And Evidence:**

Yes

**Requested Changes:**

I don't have any proposed changes. The paper is pretty straightforward.

**Strengths And Weaknesses:**

## Strengths
- The paper tackles an under-explored aspect of AI interpretability by focusing on the faithfulness of CoT reasoning, which is crucial for trusting AI systems in critical applications.
- It expands on previous research by testing across multiple model families and sizes, which enhances the generalizability of the findings.
Quantitative and Qualitative Insights:
- The combination of quantitative metrics and qualitative examples provides a robust analysis of how and why model outputs may vary with changes in CoT presentation.
- Discovering that the faithfulness metric is sensitive to the order of answer choices highlights an important aspect of model behavior that could impact applications requiring reliable reasoning.

## Weaknesses
- The generalizability of results across languages other than English and non-multiple-choice formats is not addressed, which could be important in global or varied application contexts.

---

> ### Author Response · Authors · 2024-05-14
> **Response to Reviewer H9RD**
>
> We thank Reviewer H9RD for taking the time to read and review our paper! We appreciate that the reviewer agrees with the importance of evaluating CoT faithfulness and building on the generalizability of previous evaluations.
>
> ### **Re: “The generalizability of results across languages other than English and non-multiple-choice formats is not addressed, which could be important in global or varied application contexts.”**
>
> Although we agree that the evaluation of chain-of-thought faithfulness across languages is generally important, the lack of such evaluation would be more pressing if our work were to suggest that the proposed faithfulness measurement is adequate for English. Since we argue it is not even suitable for one language for reasons that are not specific to that language, we expect it would similarly be unsuitable for other languages.
>
> We choose tasks following the experimental setup from Lanham et al. which includes eight multiple-choice benchmarks and a free-text arithmetic task (Figure 3). While we agree that adding additional task formats would provide another perspective on the metric’s generalizability, we don’t believe those experiments would contradict our findings from the MCQ benchmarks.

---

### Decision · Action_Editor_vQuM · 2024-06-10

**Recommendation:** Accept as is

**Comment:**

The reviewers unanimously recommend acceptance of the paper.  The main concerns of the reviewers were addressed by the author response and revisions of the paper.

**Audience:**

Yes: Researchers interested in LLMs.

**Claims And Evidence:**

The paper claims that Lanham et al.'s previous work on assessing unfaithfulness ignores an important confounder: the order in which choices are provided. The paper carries out experiments where the order of choices is controlled via a normalized unfaithfulness metric.  The results show that the size of the LLM does not influence unfaithfulness as suggested by Lanham et al.  The results are verified with respect to multiple LLM families.